# Inactivation Kinetics of Foodborne Pathogens in Carrot Juice by High-Pressure Processing

**DOI:** 10.3390/biology12111383

**Published:** 2023-10-29

**Authors:** Chiu-Chu Hwang, Chung-Saint Lin, Yun-Ting Hsiao, Ya-Ling Huang, Feng-Lin Yen, Yi-Chen Lee, Yung-Hsiang Tsai

**Affiliations:** 1Department of Seafood Science, National Kaohsiung University of Science and Technology, Kaohsiung 811213, Taiwan; omics1@gmail.com (C.-C.H.); cdd828@gmail.com (Y.-T.H.); ylhuang@nkust.edu.tw (Y.-L.H.); 2Department of Food Science, Yuanpei University of Medical Technology, Hsin-Chu 300102, Taiwan; chungsl@mail.ypu.edu.tw; 3Department of Fragrance and Cosmetic Science, Kaohsiung Medical University, Kaohsiung 807378, Taiwan; flyen@kmu.edu.tw

**Keywords:** high-pressure processing, carrot juice, foodborne pathogens, destruction kinetic, *Listeria monocytogenes*

## Abstract

**Simple Summary:**

Although *Listeria monocytogenes* had poor pressure resistance, its characteristic as a low-temperature tolerant bacteria allowed residual or injured bacteria by HPP to self-repair and grow during low-temperature storage. Therefore, once HPP-treated carrot juice was contaminated by *L. monocytogenes*, foodborne illness risk persisted despite refrigeration of the juice.

**Abstract:**

In this study, *Salmonella* Typhimurium, *Escherichia coli*, and *Listeria monocytogenes* were separately inoculated in sterilized carrot juice and subjected to various types of high-pressure processing (HPP) at 200–600 MPa for 0.1–15 min to observe the effects of HPP on the inactivation kinetics of foodborne pathogens in carrot juice. The first-order model fits the destruction kinetics of high pressure on foodborne pathogens during the pressure hold period. An increase in pressure from 200 to 600 MPa decreased the decimal reduction time (D values) of *S.* Typhimurium, *E. coli*, and *L. monocytogenes*. Under pressure ≥ 400 MPa, the D values of *E. coli* were significantly higher than those of *S.* Typhimurium and *L. monocytogenes*, indicating that *E. coli* had greater resistance to high pressures than the others. The Zp values (the pressure range that causes the D values to change by 90%) of *E. coli*, *S.* Typhimurium, and *L. monocytogenes* were 195, 175, and 170 MPa, respectively. These results indicated that *L. monocytogenes* and *E. coli* were the most and least sensitive, respectively, to pressure changes. Additionally, the three bacteria were separately inoculated into thermal-sterilized carrot juice and subjected to 200–600 MPa HPP for 3 min. The treated carrot juices were stored at 4 °C for 27 d. Following *S.* Typhimurium and *E. coli* inoculation, the bacterial counts of the control and 200 MPa treatments remained the same during the storage duration. However, they decreased for the 300 and 400 MPa treatment groups with increasing storage duration. During the storage period, no bacterial growth was observed in the 500 and 600 MPa treatments. However, the bacterial number for the control and pressure treatment groups increased with prolonged storage duration following inoculation with *L. monocytogenes*. Therefore, following HPP, residual *L. monocytogenes* continued growing stably at low temperatures. Overall, HPP could inhibit and delay the growth of *S.* Typhimurium and *E. coli* in carrot juice during cold storage, but it was ineffective at inhibiting the growth of *L. monocytogenes*. There was a risk of foodborne illness despite the low-temperature storage of juice. The innovation of this preliminary study is to find the impact of high pressure on the inactivate kinetics of three food pathogens in carrot juice and its practical application in simulated contaminated juice.

## 1. Introduction

Carrots and their processed products, including juice, are generally recognized as rich sources of plant nutrients, such as bioactive compounds, carotenoids, minerals, and vitamins. Notably, carotenoids, particularly α-carotene, β-carotene, and lutein, are found in abundance. Among these, β-carotene, the predominant carotenoid in carrots, serves as a precursor to vitamin A, playing a crucial role in promoting visual health. In this context, it is significant to highlight that the main carotenoid present in carrots is primarily β-carotene. Some researchers have suggested that carrots contribute to health promotion through their antioxidant and anti-inflammatory activities, along with their potential to enhance immune function [1].

In addition, carrots produce phytoalexins, which are compounds with a low molecular weight that protect plant tissues from microbial damage [2]. Research [3] concluded that phytoalexin, 6-methoxymellein, has inhibitory effects on a variety of bacteria, yeasts, and molds and is widely more effective against Gram-positive than Gram-negative bacteria. Raw, unprocessed carrot juice, as a commercial product, has short shelf storage owing to the high pH (6.4–6.8). This is because putrefactive and pathogenic microorganisms can proliferate at that pH level [4]. Therefore, the traditional processing methods of thermal sterilization or pasteurization at relatively high temperatures (70–90 °C) for 1–6 min must be applied when manufacturing carrot juice for commercial sale. However, the quality and nutritional value of the juice are negatively affected by thermal processing [5].

Non-thermal processing techniques have been developed rapidly over the past few decades as alternative methods to overcome the disadvantages of thermal treatments. Among these, high-pressure processing (HPP) is an excellent non-thermal processing technique. The pressurization range of 200–600 MPa is employed to inactivate putrefactive and pathogenic bacteria, thereby, extending the food shelf life and improving food safety [6]. Hence, HPP has been employed in fresh fruit and vegetable juices to inactivate foodborne pathogens without, or minimally, affecting their nutritional and quality attributes [7]. Teo et al. [8] concluded that after 615 MPa was applied for 2 min, the counts of *Escherichia coli* and *Salmonella* spp. inoculated into carrot juice were reduced by approximately 6.4 and 5.1–7.8 logs, respectively. Moreover, 500 MPa HPP for 2 min reduced *Listeria innocua* and *E. coli* in carrot juice by roughly 4 and 5 log CFU/mL, respectively [9].

Microorganisms’ resistance to HPP is influenced by multiple intrinsic and environmental factors, particularly the nature of the medium. When bacteria exist in nutrient-rich culture media, their viability under HPP substantially improves. This is because the culture media contains substances that provide the bacteria protection from damage or essential nutrients for restoration [10]. Research [11] demonstrated that the resistance of foodborne pathogens to HPP increased with reductions in water activity. Patterson [12] reported in an article that some food components, including proteins, carbohydrates, and lipids, protect the microbes. To improve the safety and stability of HPP for food processing, the initial microbial population must be reduced to a satisfactory level by treatment. Therefore, it is necessary to perform kinetic analysis for specific pathogenic bacteria when HPP is applied and ascertain the pressure dependence at which the bacterial inactivation rate is reached [13]. According to the microbiological hygiene standards announced by the Taiwan Food and Drug Administration, *E. coli* and *Salmonella* must not be detected in beverages termed fruit juices. *Listeria monocytogenes* is a low-temperature tolerant bacteria that can survive and proliferate during refrigeration. It is a potential pathogen in refrigerated ready-to-eat foods, especially low-acidity juices [14]. The United States establishes microbial regulations for juices under the authority of the Food and Drug Administration (FDA) to ensure the quality and safety of food products. As per FDA guidelines, juices are not permitted to contain any pathogenic microorganisms, such as *Salmonella* spp. and *E. coli*, in order to safeguard food safety [15]. The European Union (EU) mandates that juices must not contain any pathogenic microorganisms, such as *Salmonella* spp. and *E. coli*, to assure the safety of the food product [16].

Given the lack of information on HPP’s inactivation kinetics on the aforementioned pathogenic microorganisms in fruit juices, this work intended to determine the inactivation kinetics of high pressure on *E. coli*, *S.* Typhimurium, and *L. monocytogenes*. Additionally, HPP’s effects on the viability of pathogenic bacteria in juice samples during refrigeration were simulated following the contamination of the carrot juice by the three pathogenic strains.

## 2. Materials and Methods

### 2.1. Preparation of Sterilized Carrot Juice

To prepare the juice, 10 kg of carrots (*Daucus carota* L.) were purchased from a traditional vegetable market, washed with tap water, peeled, and cut into pieces (approximately 3.5 cm in diameter and 0.5 cm in thickness). The carrot pieces were mixed with deionized water at 1:3 (*w*/*w*) and homogenized in a homogenizer. After the homogeneous solution was filtered through a double-layered fine cloth, the filtrate was bottled into 1.0 L serum bottles for sterilization inside an autoclave (TM-329, Tomin Medical Equipment Co., Ltd., New Taipei city, Taiwan) at 121 °C for 15 min. The purpose of sterilization was to kill the original microorganisms in carrot juice. The solutions were cooled to room temperature and refrigerated for later use. Figure 1 shows the flow diagram for experiment design.

### 2.2. Preparation of Bacterial Culture Medium

Three strains of foodborne pathogens were used in this experiment—*E. coli* (ATCC 25922), *S.* Typhimurium (BCRC12947), and *L. monocytogenes* (ATCC 19115). These were obtained from the Bioresource Collection and Research Center (Hsinchu, Taiwan). One loop of each strain was inoculated into 5 mL of trypticase soy broth (TSB, Difco, BD, Sparks, MD) and cultured in a shaking incubator at 30 °C (25 °C for *L. monocytogenes*) for 12–18 h. Since *L. monocytogenes* is a low-temperature tolerant bacterium and the optimal growth temperature of this strain used in this study is about 25 °C, the subsequent culture temperature of *L. monocytogenes* was set at 25 °C [17]. Next, 0.1 mL of the bacterial solution was added to 100 mL of TSB and shaken for 12–18 h before being centrifuged at 7000× *g* at 4 °C for 8 min. The supernatant was discarded. Following that, 10 mL of sterilized phosphate buffer solution (PBS, pH 7.4) was put into the previous solution and mixed well before being centrifuged again (7000× *g* at 4 °C for 8 min). The supernatant was removed, and the precipitated bacteria was suspended in sterilized PBS.

The absorbance of the suspended bacterial solution was measured using a spectrophotometer at a wavelength of 600 nm until the value was 2.6–2.7. At this time, the bacterial count was approximately 9.0 log CFU/mL. After diluting the suspensions in sequence with sterilized PBS, 0.1 mL of the serialized dilution was smeared onto the trypticase soy agar (TSA) medium and cultured at 30 °C for 24 h (*L. monocytogenes* was cultured at 25 °C for 48 h). The total colonies on each Petri dish were counted to confirm the actual bacterial concentrations.

### 2.3. Processing of Inoculated Carrot Juice

The sterilized carrot juice was divided into 150 mL polyethylene terephthalate (PET) bottles that were sterilized with 75% ethanol. Next, 1 mL of the *S*. Typhimurium, *E. coli*, and *L. monocytogenes* were separately inoculated into the PET-bottled carrot juice. The inoculum of the juice samples at this stage was roughly 7.0–8.0 log CFU/mL. After the carrot juice samples were inoculated with each strain of foodborne pathogenic bacteria, they were divided into two batches. The first batch was used for analyzing the inactivation kinetics of microorganisms by HPP. The second batch was used to simulate variations in the bacterial counts of contaminated juice samples after HPP and during refrigeration. Each experiment treatment was conducted with three independent bottles. The data for each pressurized condition are expressed as the average of triplicates.

### 2.4. HPP Conditions

The juice samples were set in an HPP device (Bao Tou KeFa, Inner Mongolia, China) that was 200 mm in diameter and depth. It had a capacity of 6.2 L and a working pressurization range of 0.1–600 MPa. An aqueous solution at room temperature (23 °C) was regarded as the medium for pressurization transmission. The mean rate of temperature rising in the pressurized solution was 2.0 ± 0.5 °C per 100 MPa increasing pressure. The maximum pressure was reached within 1.6 min and decompression occurred over 12–16 s. The HPP conditions were applied in two parts. In the first part, the kinetics of killing pathogenic bacteria was examined by treating the inoculated samples at 200, 300, 400, 500, and 600 MPa and pressurizing for various durations (0.1–15 min). Subsequently, the residual bacterial count was determined. In the second part, the inoculated samples were subjected to 200, 300, 400, 500, and 600 MPa for 3 min before cold storage. Changes in the surviving bacterial count were observed over 27 days. All treatments at the same pressure and time point were conducted in triplicate. The control group (without HPP) was placed under atmospheric pressure (0.1 MPa).

### 2.5. Detection of Bacterial Counts

To test for the inactivation kinetics of bacteria, 1 mL of carrot juice sample was put into a tube containing 9 mL of sterilized physiological saline solution. After diluting in sequence, 0.1 mL of the various dilutions (in duplicates) were taken and smeared evenly onto the TSA medium and incubated at 30 °C for 24 h (*L. monocytogenes* was cultured at 25 °C for 48 h). The bacterial colonies were subsequently counted. To simulate contaminated carrot juice samples, the HPP samples were stored in refrigerator for 27 days. Samples were retrieved every three days for the detection of bacterial counts. All the data were expressed in terms of the mean ± standard deviation derived from three independent experiments.

### 2.6. Determination of the D and Zp Values

The pressurization destruction kinetics of foodborne pathogens in fruit juice during the stage of pressure x pressurization duration was analyzed to reveal a death logarithmic level of the first-order reaction. The results were expressed as log (N/N0) = −kt where N is the surviving pathogenic bacterial count after the pressurization duration t (min), N0 is the initial bacterial count before HPP, and k is a constant for the reaction rate (min*^−^*^1^). The D value is the duration required to reduce the pathogenic bacterial count by 90% at a specific pressure. It is obtained using the negative reciprocal of the slope of the survival curve for log (N/N0) against time (excluding the untreated control group) and is expressed as D = −(1/slope). The pressure sensitivity of the D value is determined by plotting the D value against the decimal logarithm of the pressure. The Zp value (z-value of pressure) is the negative reciprocal of the slope, expressed as Zp = −(1/slope). Zp is defined as the amount of increased pressure required to change the D value by 90%.

### 2.7. Statistical Analysis

All values of this study were employed for an analysis of variance (ANOVA) of the various treatment conditions and reported as the average ± standard deviation obtained from triplicate samples. The statistical software SPSS version 12.0 (St. Armonk, New York, USA), Tukey test, and one-way ANOVA were used for statistical analysis with *p* < 0.05 exhibiting a significant difference.

## 3. Results

### 3.1. Inactivation Kinetics of HPP on Foodborne Pathogens in Carrot Juice

Figure 2 shows the bacterial survival rates after the carrot juice was separately inoculated with approximately 10^8^ CFU/mL of *S.* Typhimurium, *E. coli*, and *L. monocytogenes* before subjection to various pressures (200–600 MPa) and pressurization durations (0.1–15 min). The bacterial counts gradually decreased with increasing pressures and durations. The duration required for the count to decrease was shorter with increased pressure. The survival curves at the high-pressure levels were steeper than those at the low-pressure levels, confirming the greater destruction rate of the high-pressure levels. The data for these curves fitted a first-order model, indicating that pressure destruction of the pathogenic bacteria strains in the fruit juice during the pressurization duration conformed to a semi-logarithmic model.

The D value calculated from the survival curve of an organism could be used to compare the resistance of microbes and the effectiveness of HPP. The D values of all three pathogens decreased with increasing pressure. When pressurization increased from 200 to 600 MPa, the D values of *S.* Typhimurium decreased from 9.55 to 0.07 min, *E. coli* from 10.49 to 0.10 min, and *L. monocytogenes* from 11.02 to 0.03 min. At lower pressures (200–300 MPa), the D values of the three pathogenic bacteria differed although the variations were small. However, the *E. coli* treated with 400, 500, and 600 MPa in carrot juice had higher D values of 0.91 min, 0.19 min, and 0.10 min, respectively, than *S.* Typhimurium (0.42 min, 0.16 min, and 0.07 min, respectively), and *L. monocytogenes* (0.39 min, 0.16 min, and 0.03 min, respectively). When the pressure was ≥400 MPa, the D values of *E. coli* were significantly higher than those of *S.* Typhimurium and *L. monocytogenes*. This inferred that *E. coli* was more tolerant to higher pressures than the other two strains, and *L. monocytogenes* was most sensitive to pressure treatment compared with *S.* Typhimurium and *E. coli*.

The decimal reduction time curve of the HPP derived from plotting the decimal logarithm of the D value against pressurization is presented in Figure 3. The Zp values of *E. coli*, *S.* Typhimurium, and *L. monocytogenes* were 195, 175, and 170 MPa, respectively. The Zp value analyses indicated bacterial sensitivity to pressure changes. Based on the destruction rates, *L. monocytogenes* was the most sensitive to pressure changes, followed by *S.* Typhimurium. *E. coli* was the least sensitive. This also shows that in food processing using HPP treatment, *E. coli* would require longer holding times to achieve the same damaging effect as *S.* Typhimurium and *L. monocytogenes*. In carrot juice pasteurization, if pathogen inactivation becomes a requirement, *E. coli* would become a more appropriate target pathogen.

### 3.2. Effects of HPP on Bacterial Count in Carrot Juice during Cold Storage

The residual bacterial counts in the carrot juice after treatment at various pressures for 3 min are presented in Table 1. There was no obvious change for the 200 MPa treatment compared with the control, except for a slight decrease in the count of *L. monocytogenes*. The microbial counts in 400 Mpa treatment groups decreased significantly with pressure increased (*p* < 0.05). No residual bacteria were detected in the 500 and 600 Mpa treatments. The residual *E. coli* number (5.82 log CFU/mL) in the 400 Mpa treatment was obviously higher than that of *S.* Typhimurium (2.28 log CFU/mL) and *L. monocytogenes* (1.36 log CFU/mL) (*p* < 0.05). This result correlated with the conclusion on the D values stated in the earlier sections, that is, *E. coli* had the greatest pressure resistance.

Figure 4 presents the changes in bacterial counts in carrot juice inoculated with *S.* Typhimurium after HPP and storage in the refrigerator. The bacterial numbers of the control and 200 MPa treatments were unchanged during storage (27 d). *Salmonella* Typhimurium counts varied between 7.2 and 8.0 Log CFU/mL throughout the entire storage period. For the 300 MPa treatment group, the bacterial count (4.86 Log CFU/mL) decreased slowly at the start of the storage period (before the sixth day) and reached 2.50 Log CFU/mL by the storage of day 6; however, it remained the same thereafter, without any significant change. The residual bacterial count was obviously lower than that of the control and 200 MPa treatments. The bacterial level (2.28 Log CFU/mL) of 400 MPa treatment continually decreased during the initial stage of storage. No bacterial count could be detected after the sixth day. Similarly, no bacterial count was detected during storage for the high-pressure treatment groups (500–600 MPa), indicating that the higher the pressure, the greater the effectiveness of the inhibition of *S.* Typhimurium growth. This study showed that when carrot juice was contaminated with *S.* Typhimurium, it was possible to inhibit the growth of microorganisms in the juice during refrigeration by using a pressure treatment of at least 300 MPa for 3 min.

Changes in bacterial counts in carrot juice inoculated with *E. coli* after HPP and kept in the refrigerator are presented in Figure 5. During storage (27 d), the bacterial counts of the control and 200 MPa treatments did not change much. *Escherichia coli* counts varied between 7.70 and 8.20 Log CFU/mL throughout the entire storage period. For the 300 MPa treatment group, the bacterial count (7.40 Log CFU/mL) decreased slowly during the storage period and reached 6.90 Log CFU/mL by the end of storage (day 27). However, it slowly decreased for the 300 MPa treatment group, with the residual bacterial count being significantly lower than that of the control and 200 MPa treatments. For the 400 MPa treatment, the bacterial counts decreased continuously at the beginning of storage and reached the lowest count on day 18. Subsequently, the bacterial counts increased slightly and reached 2.23 Log CFU/mL by the end of storage (day 27). The bacterial counts of the *E. coli*-inoculated carrot juices subjected to pressurization at 400 MPa were significantly lower than those of the control, 200 MPa, and 300 MPa over the entire storage duration. No bacterial count was detected in the higher pressure treatment groups (500 and 600 MPa) during storage. These results showed that the higher the pressure, the more effectively the growth of *E. coli* was inhibited in carrot juice within cold storage. This study also found that when carrot juice was contaminated with *E. coli*, it was possible to inhibit the growth of microorganisms in the juice during refrigeration by using a pressure treatment of at least 400 MPa for 3 min.

Figure 6 presents the changes in bacterial counts in carrot juice inoculated with *L. monocytogenes* following HPP and kept at 4 °C. The bacterial count in the control and 200 MPa treatment groups increased rapidly with longer storage durations. Because the bacterial count was too high and caused sample spoilage, the experiments were terminated after the 15th and 21st days for the control and 200 MPa treatment groups, respectively. Similar changes were observed in the 300 MPa treatment. However, the bacterial counts of 300 MPa after day 9 were obviously lower than those of the control and 200 MPa treatments for the same storage duration (*p* < 0.05). In the 400 MPa group, the *L. monocytogenes* count in carrot juice increased rapidly during the first 6 days of storage and gradually thereafter; however, the bacterial counts remained significantly lower than those of the control, 200 MPa, and 300 MPa groups within the entire storage period. Bacterial growth was detected on the sixth and ninth day for the 500 and 600 MPa treatment groups, respectively. Subsequently, the bacterial counts for the 500 and 600 MPa treatment groups continued to increase slowly and reached 7.25 and 5.00 Log CFU/mL, respectively, by the end of storage duration (day 27). Overall, it can be seen that *L. monocytogenes* can reliably proliferate at refrigerated temperatures, especially in nutrient-rich carrot juice. This is because *L. monocytogenes* is a low-temperature tolerant bacterium that can continue to grow under refrigerated conditions, even after pressure treatment [18]. This study showed that when carrot juice was contaminated with *L. monocytogenes*, although higher pressure (400–600 MPa) will significantly reduce the number of bacteria or reach undetectable levels, it will continue to grow during storage in refrigeration.

## 4. Discussion

### 4.1. Study on Inactivation Kinetics of HPP on Foodborne Pathogens in Carrot Juice

The D values obtained from the survival curve could be employed to compare the resistance of various bacteria and HPP’s effectiveness. Figure 1 presents the slopes, D values, and linear regression coefficients of the pathogenic bacteria strains in sterilized carrot juice after various HPPs (200–600 MPa, 0.1–15 min). The D value represents the time required to decrease the bacterial count by a log value—the greater the D value, the longer it takes to decrease a certain microorganism count [19]. Smelt and Rijke [20] stated that after being subjected to high-pressure inactivation, *E. coli* presented first-order kinetics in a physiological saline solution. Its D values for treatments with 200, 250, 300, and 350 MPa were 25.9, 8.0, 2.5, and 0.8 min, respectively. Research [21] concluded that under 700 MPa, *E. coli* in PBS had a D value of approximately 13 min. Simpson and Gilmour [10] demonstrated the inactivation of *L. monocytogenes* in PBS, with D-values of 3.7, 1.6, and 0.6 min for HPP treatments at 400, 500, and 600 MPa, respectively. However, the *L. monocytogenes* at 350 MPa had about 5.1 min of D-value in fish homogenate [19]. The different D values of *E. coli* or *L. monocytogenes* obtained from various HPPs could be attributed to the varying environmental media and bacterial strains [22,23].

This study found that *E. coli* and *S*. Typhimurium (Gram-negative bacteria) showed greater resistance than *L. monocytogenes* (Gram-positive bacteria) under high pressure. In general, *E. coli* exhibited greater tolerance to high pressures when foodborne pathogens in carrot juice were subjected to HPP [9]. However, Gram-positive bacteria are considered to be more resistant to HPP than Gram-negative bacteria [13]. Previous research has shown that food components can protect microorganisms from the effects of HPP [10]. In addition, changes in microbial sensitivity to HPP may be due to many other interacting factors in the food itself [12]. Studies have shown that carrot juice has protective effects on *E. coli* [24]. Another study suggested that the phytoalexins in carrots are less toxic to *E. coli* [25], which may make this strain more pressure-resistant in carrot juice. Kurosaki and Nishi [26] found that the phytoalexin, 6-methoxymellein, in carrots, was more effective at inhibiting Gram-positive than Gram-negative bacteria. Since carrot juice has a significant inhibitory effect on Gram-positive bacteria, especially *L. monocytogenes* [27], the strain was less resistant to HPP.

### 4.2. Study on Effects of HPP on Bacterial Count in Carrot Juice during Cold Storage

After *S.* Typhimurium and *E. coli* were treated with 300 and 400 MPa, the bacterial counts in the initial storage period continued to decrease, while their bacterial counts of 500 and 600 MPa groups did not grow during storage. This was because the high pressures damaged the bacterial cells and rendered them incapable of resisting low temperatures, causing them to die [28]. These study results were similar to Patterson et al. [14] who studied *E. coli* inoculated into irradiation-sterilized carrot juice, kept in a refrigerator for 14 d. They demonstrated that the bacterial count in the control group (without HPP) remained unchanged. However, the bacterial level of the HPP group (500 MPa/1 min) decreased continuously during storage and reached untraceable levels by day 10.

In this study, although the bacterial count of *L. monocytogenes* reduced or became undetectable after high-pressure treatment (400–600 MPa), its growth recovered significantly and rapidly during cold storage. This may be due to some sub-lethally injured bacteria having self-repaired the damages resulting from HPP, followed by overgrowth during storage [29]. *Listeria monocytogenes* is a low-temperature tolerant bacteria and a storage temperature of 4 °C was a suitable growth condition for the sub-lethally injured pathogens. Therefore, special attention must be paid to the fact that although *L. monocytogenes* has a low tolerance to high pressures, it is a low-temperature tolerant bacteria. This means that when its cells are damaged by pressure, they can self-repair in low-temperature environments, thereby, enabling the pathogen to proliferate [29]. Overall, once carrot juice is contaminated with *L. monocytogenes*, the risk of food poisoning persists even when it is stored under low temperatures.

In contrast, Patterson et al. [14] highlighted that after *L. monocytogenes* was added to irradiation-sterilized carrot juice and subjected to pressurization (500 MPa/1 min), no bacteria were subsequently detected (>6 log reduction). Furthermore, no bacterial growth was observed within subsequent stores at 4 °C, 8 °C, or 12 °C for 14 d. Their study demonstrated a synergistic effect between the antilisterial characteristics of phytoalexins in carrot juice and high-pressure treatment [14]. When *Listeria* was inoculated into thermal-treated carrot juice, the bacterial count gradually increased during cold storage, confirming that thermal treatment may reduce the inhibitory effect. According to Beuchat and Brackett [27], thermal treatment reduces the antibacterial properties of carrot juice. Therefore, the results of our study differed from those reported by Patterson et al. [14]. A possible reason could be that the thermal sterilization treatments decreased antilisterial activities in the carrot juice; hence, *Listeria* growth could not be inhibited during cold storage. Therefore, non-thermal sterilized juice should be used as a matrix for bacterial inoculation in future studies.

When bacteria are treated with high pressure, the permeability of the bacterial cell membrane is destroyed, followed by loss of membrane integrity, cell swelling, and finally cell death. When observing bacterial cells after HPP with an electron microscope, it was found that microbial cells showed varying degrees of resistance to HPP treatment [30]. Furthermore, the changes in cell morphological characteristics or intracellular enzyme activity caused by pressurization vary greatly depending on the bacterial genus [30]. Wang et al. [31] found that the main mechanism of high-pressure inactivation of bacteria is that pressure treatment destroys the proteins in the cells. In addition, high-pressure treatment causes varying degrees of damage to different types of bacterial cells, including damaging cell walls, releasing cytoplasmic contents, and changing cell membrane permeability, as well as causing other biochemical changes [31]. In this study, the mechanisms responsible for the inactivation of *S.* Typhimurium, *E. coli*, and *L. monocytogenes* may be related to cell membrane disruption, cell wall rupture, and leakage of cellular contents.

There are many studies confirming that the ingredients originally contained in carrot juice have inhibitory effects on certain bacteria, in particular *L. monocytogenes* [27,32]. Within the realm of food processing, a diverse range of cutting-edge technologies is currently being harnessed for enhancing fruit juice processing, thereby elevating product quality, safety, and nutritional value. For instance, pulsed electric field (PEF) technology employs brief yet potent electric fields to disrupt cell membranes and deactivate microorganisms. This innovative approach effectively maintains the juice’s innate color and nutritional profile. Conversely, ultrasonic treatment employs vibrations to generate and rupture minuscule bubbles within the liquid, generating localized high-energy outputs that physically dismantle microorganisms. Furthermore, high-pressure homogenization involves subjecting juice to elevated pressure to mechanically disintegrate microbial cells, effectively reducing their presence. Each of these pioneering techniques boasts distinctive advantages and limitations, offering the adaptability to tailor solutions to the unique characteristics and demands of the juice. Notably, meticulous assessment and rigorous pre-application testing of each method are of paramount importance.

Compared with conventional thermal sterilization processing, carrot juice treated with non-thermal processing technology, HPP, can maintain its natural nutrients, flavor, and color [4,31]. In the meantime, high-pressure processing juice is also taken as a “clean label food” because it is without the addition of preservatives and food additives [4,31]. Recently, because people’s requirements for natural, nutritious, and healthy food are gradually increasing, high-pressure juice has been widely used commercially, and the world’s output value has exceeded USD 10 billion [31]. Therefore, this result in my work would be beneficial to the food industry in developing safe processing conditions for HPP-treated carrot juice. Since residual or injured *L. monocytogenes* after high-pressure treatment can continue to survive and grow under refrigerated conditions, further work is needed to address this issue to ensure food safety. Overall, the innovation of this research is to find the impact of high pressure on the inactivate kinetics of three food pathogens in carrot juice and its practical application in simulated contaminated juice.

## 5. Conclusions

This is a preliminary study to indicate that *E. coli* was more tolerant to high pressures than *S.* Typhimurium and *L. monocytogenes*. *Listeria monocytogenes* was the most sensitive to pressure change. In addition, HPP (≥400 MPa) could effectively inhibit and delay the growth of *E. coli* and *S.* Typhimurium in carrot juice; however, *L. monocytogenes* in HPP-treated juice continued to grow during storage at 4 °C. To summarize, although *L. monocytogenes* had poor pressure resistance, its characteristic as a low-temperature tolerant bacteria allowed residual or injured bacteria to self-repair and grow during low-temperature storage. Therefore, once HPP-treated carrot juice was contaminated by *L. monocytogenes*, food poisoning risk persisted despite refrigeration of the juice. The study of this inactivation kinetic and cold storage test of bacterial inoculation in carrot juice would be very beneficial to the food industry by developing safe processing conditions for HPP-treated foods.

## Figures and Tables

**Figure 1 biology-12-01383-f001:**
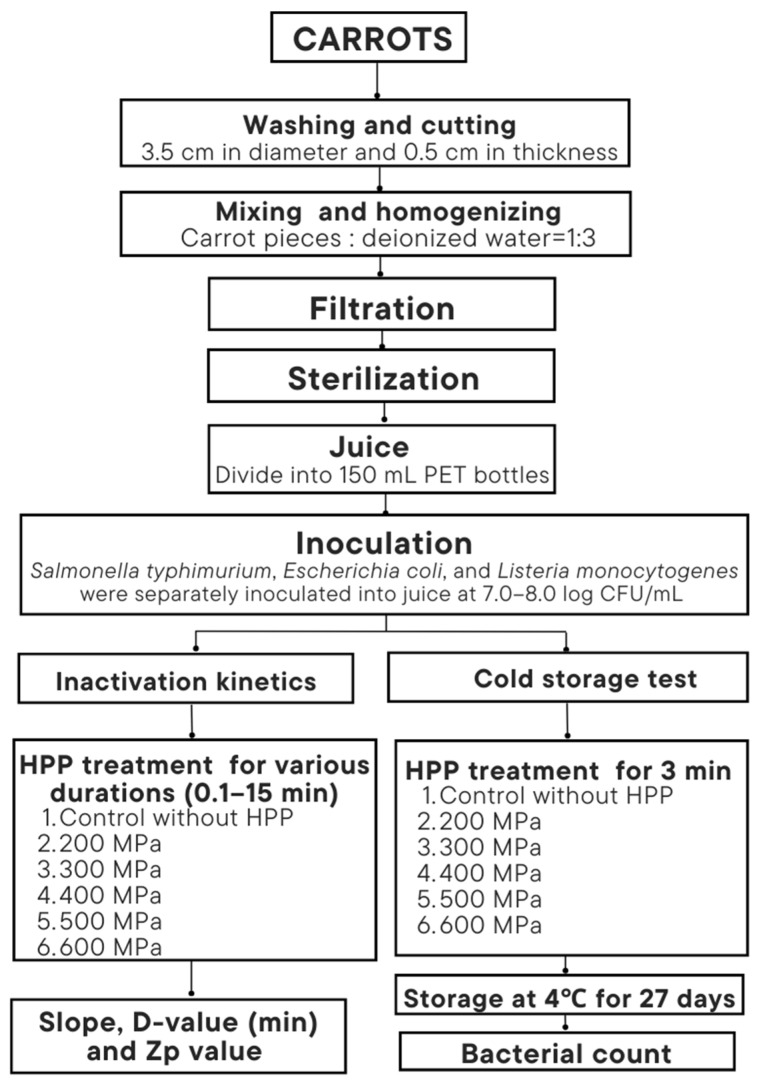
Experiment design.

**Figure 2 biology-12-01383-f002:**
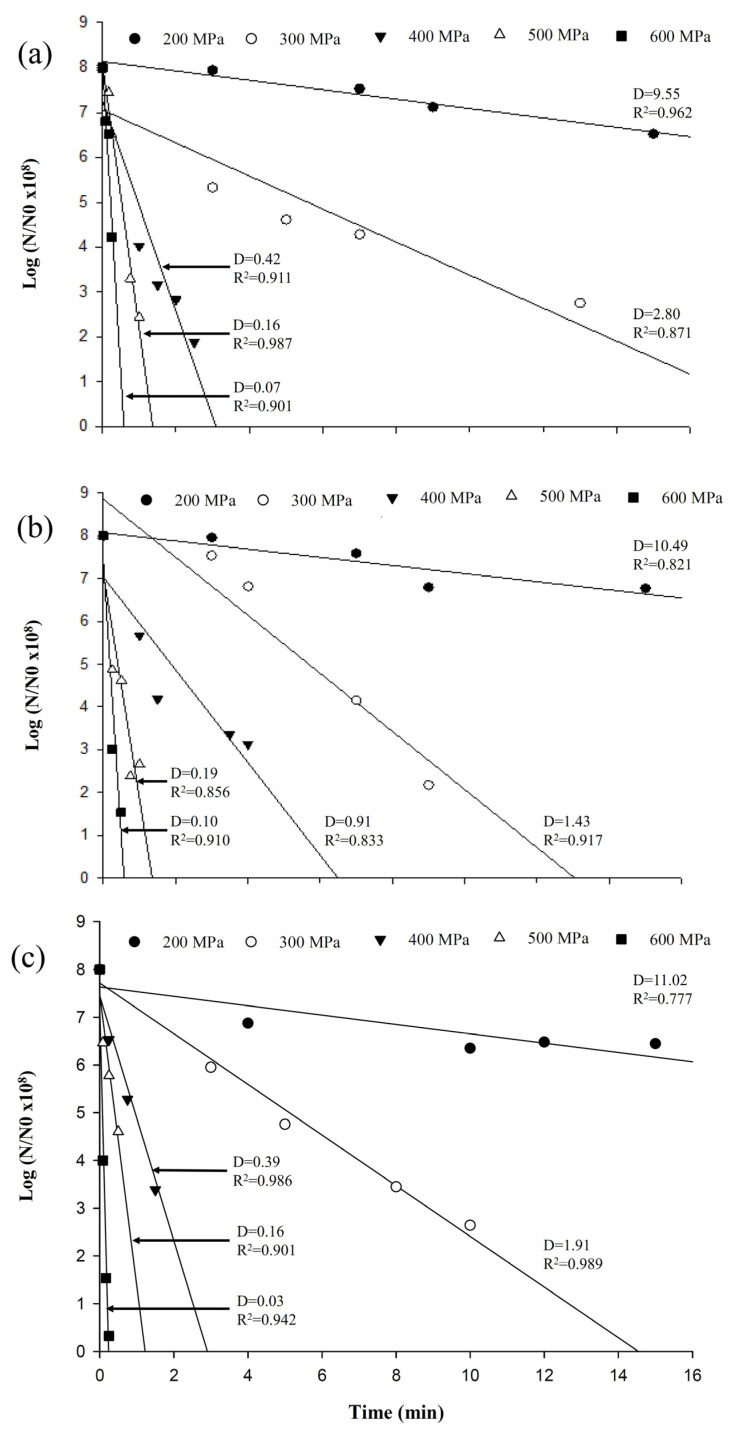
HPP survival curves, the D values and R^2^ values of *Salmonella* Typhimurium (**a**), *Escherichia coli* (**b**), and *Listeria monocytogenes* (**c**) in sterilized carrot juice with different pressurization (200–600 MPa) and times (0.1–15 min).

**Figure 3 biology-12-01383-f003:**
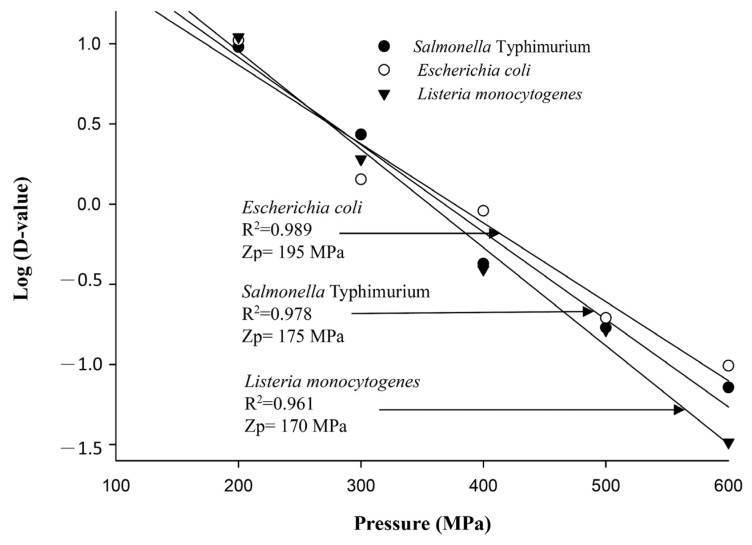
HPP decimal reduction time curves for *Salmonella* Typhimurium, *Escherichia coli*, and *Listeria monocytogenes* in sterilized carrot juice.

**Figure 4 biology-12-01383-f004:**
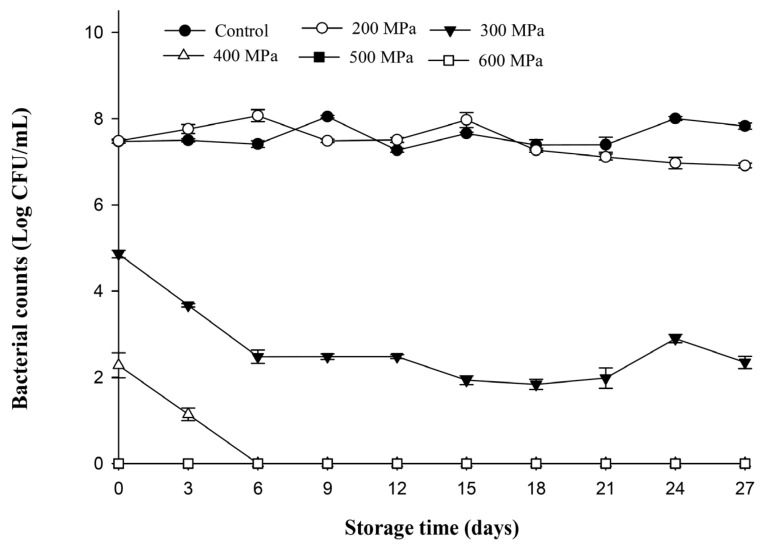
Changes in *Salmonella* Typhimurium inoculated in sterilized carrot juice after high-pressure treatments (200, 300, 400, 500, and 600 MPa for 3 min) and kept at 4 °C for 27 days. All data were the average ± standard deviation of three independent samples (*n* = 3).

**Figure 5 biology-12-01383-f005:**
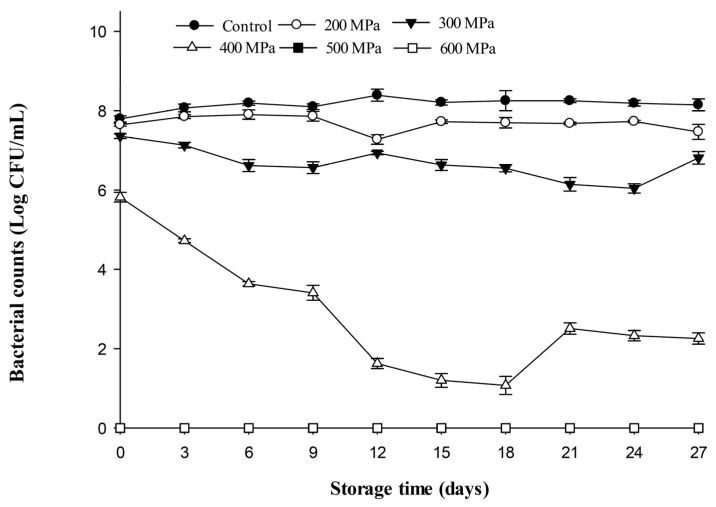
Changes in *Escherichia coli* inoculated in sterilized carrot juice after high-pressure treatments (200, 300, 400, 500, and 600 MPa for 3 min) and kept at 4 °C for 27 days. All data were the average ± standard deviation of three independent samples (*n* = 3).

**Figure 6 biology-12-01383-f006:**
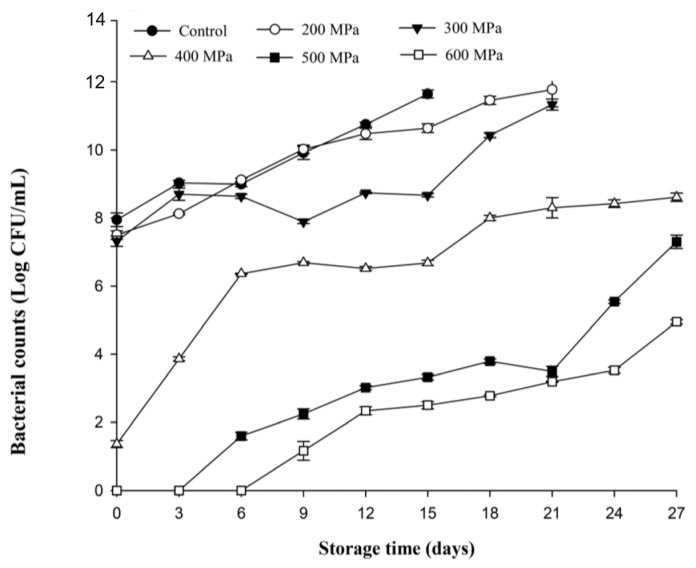
Changes in *Listeria monocytogenes* inoculated in sterilized carrot juice after high-pressure treatments (200, 300, 400, 500, and 600 MPa for 3 min) and kept at 4 °C for 27 days. All data were the average ± standard deviation of three independent samples (*n* = 3).

**Table 1 biology-12-01383-t001:** The survival of *Salmonella* Typhimurium, *Escherichia coli,* and *Listeria monocytogenes* inoculated into carrot juice after high-pressure treatments (200, 300, 400, 500, and 600 MPa for 3 min).

	Survival Plate Count (Log CFU/mL)					
	Control (Inoculated)	200 MPa	300 MPa	400 MPa	500 MPa	600 MPa
*Salmonella* Typhimurium	7.47 ± 0.03 ^aA1^	7.48 ± 0.04 ^aA^	4.86 ± 0.09 ^bB^	2.28 ± 0.29 ^bC^	ND ^2^	ND
*Escherichia coli*	7.79 ± 0.08 ^bA^	7.65 ± 0.03 ^bA^	7.36 ± 0.06 ^aB^	5.82 ± 0.12 ^aC^	ND	ND
*Listeria monocytogenes*	7.95 ± 0.20 ^bA^	7.50 ± 0.12 ^aB^	7.32 ± 0.15 ^aB^	1.36 ± 0.10 ^cC^	ND	ND

^1^ The data were the average ± standard deviation of three independent samples: a–c: different letters within the same column point out significantly different (*p* < 0.05); A–C: different letters within the same row point out significantly different (*p* < 0.05). ^2^ ND: not detected (plate count less than 1.0 log CFU/mL).

## Data Availability

Data shown in the work is available on request from the corresponding author.

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
