# Peer review of "Inactivation Kinetics of Foodborne Pathogens in Carrot Juice by High-Pressure Processing"

_biology, 2023, doi:10.3390/biology12111383_

Round 1

Reviewer 1 Report

Comments and Suggestions for Authors

It was found that the study lacks novelty as it primarily confirms existing knowledge on the inactivation kinetics of foodborne pathogens using high-pressure processing. The work does not substantially contribute to the existing body of literature in this field.

The manuscript's scope is relatively narrow, focusing solely on carrot juice and its pathogen inactivation kinetics. Reviewers expressed concerns about the broader significance of the findings in the context of food safety and high-pressure processing applications.

The presentation of results and data visualization could be improved to enhance the clarity and interpretability of the findings. Additionally, some key results are not supported with sufficient statistical analysis.

The manuscript does not discuss the practical implications of the research findings, which could limit its relevance to food industry professionals and researchers in the field.

A comparative analysis with alternative food preservation techniques would strengthen the manuscript and provide a more comprehensive understanding of high-pressure processing's efficacy.

Serious concerns about the methodology employed, suggesting that further validation and optimization of experimental procedures would be necessary to ensure the accuracy and reproducibility of the results.

Comments on the Quality of English Language

It was found that the study lacks novelty as it primarily confirms existing knowledge on the inactivation kinetics of foodborne pathogens using high-pressure processing. The work does not substantially contribute to the existing body of literature in this field.

The manuscript's scope is relatively narrow, focusing solely on carrot juice and its pathogen inactivation kinetics. Reviewers expressed concerns about the broader significance of the findings in the context of food safety and high-pressure processing applications.

The presentation of results and data visualization could be improved to enhance the clarity and interpretability of the findings. Additionally, some key results are not supported with sufficient statistical analysis.

The manuscript does not discuss the practical implications of the research findings, which could limit its relevance to food industry professionals and researchers in the field.

A comparative analysis with alternative food preservation techniques would strengthen the manuscript and provide a more comprehensive understanding of high-pressure processing's efficacy.

Serious concerns about the methodology employed, suggesting that further validation and optimization of experimental procedures would be necessary to ensure the accuracy and reproducibility of the results.

Reviewer 2 Report

Comments and Suggestions for Authors

Dear Authors,

Thank you for this interesting work. Please find attached tow minor recommendations to be corrected in the paper if possible.

Kind regards

Reviewer 3 Report

Comments and Suggestions for Authors

line 80-82 - The authors should also cite EU and FDA microbiological requirements to the bacteria in question. The requirements are readily available and it should be remembered that the readers of the journal come from different countries and continents and not only from Taiwan.

The entire introduction was well and concretely written. The problem is outlined, the state of the knowledge is given, and a clear aim and scope of the work is given.

section 2.3. - Please indicate here as in the abstract that separate inoculations of the juices were made.

line 138- please list specifically how long the samples were kept under different strengh of pressure. (the same for line 18). Were they also held 0 min? This is more like a control sample.

line 183-185- Did the authors check the fit with other models? It would be useful to provide a table with the results of fitting to other models, e.g., zero or second-order model.

section 3.1. - The authors indicate which bacteria is more or less resistant to HPP treatment but there is a lack of in-depth analysis of why this is so and what mechanisms are behind it. This information can be taken from the literature.

line 256- according to table 2 not all bacteria decreased significantly after 300 MPa HPP processing, please verify

line 343- Based on Figures 3, 4  I would disagree that 300 MPa can successfully retard the growth of the test microorganisms. Rather, such an effect was achieved at 400 MPa.

Comments on the Quality of English Language

The entire article needs language corrections, especially the Conclusions section.

Reviewer 4 Report

Comments and Suggestions for Authors

The research topic is of great practical and scientific importance. Carrot juice is an important component of the diet, a source of vitamins. The death of bacteria in it under different temperature conditions has been poorly studied. There are no similar studies in the literature.

Despite the generally high scientific level of the manuscript, it contains many shortcomings that need to be corrected.

1. The abstract is oversaturated with numbers: I recommend leaving only the most important of them.

2. Line 16, 23, 25, 30, 37, 86, 237, 242 and many others: the italics are lost and Typhimurium is written with a small letter. The same mistake on the image of many drawings.

3. Line 31, 33 and others: remove the hyphen.

4. Line 43, 44: carotene is also an antioxidant; it is better not to list these terms separated by commas. A very superficial description. Describe more fully.

5. Line 210: The figure is not very informative: only two graphs are normally visible (for 200 and 300 MPa). I recommend making the drawing in the form of five diagrams (for each pressure), each of which will contain three graphs (for each type of bacteria). In addition, I recommend adding a linear regression equation with a determination coefficient (R2) to the chart for each trend line. Accordingly, Table 1 can be deleted. There is no need to repeat the same legend in all parts of the figure. Leave it only in the top picture.

6. Line 174: Results and Discussion are better separated. The Results should not contain references to the literature, interpretation of data, repetition of suggestions about research methods.

7. Line 263: the correctness of the statistical processing is questionable. In the authors, 7.47 + - 0.03 and 7.79 + - + 0.08 do not have significant differences. Or 7.48 + - 0.04 and 7.65 + - 0.03 also do not differ significantly. This is highly doubtful. Please clarify.

8. Line 279, 301, 312: In the title of the figure, indicate the repetition and what the vertical line means (standard deviation?).

9. Line 312: The y-axis should be extended to 14 and the legend should be moved to the top left corner of the drawing.

10. Line 338: There needs to be a deeper discussion of the results. The authors really did a lot of research, the results are new. However, the discussion of the results is incomplete. Discussion spoils the whole impression of the article. We need to look at the issue under study in a broader context.

Round 2

Reviewer 1 Report

Comments and Suggestions for Authors

Comments not addressed properly

Comments on the Quality of English Language

Why the editor sent this manuscript to me, although I already pointed out serious flaws and recommended rejection? 

Reviewer 3 Report

Comments and Suggestions for Authors

The authors addressed all comments and remarks. Missing information was completed and the discussion of the results was expanded. In my opinion, the authors have also improved the article linguistically.

Reviewer 4 Report

Comments and Suggestions for Authors

Almost all fixes have been made. But I cannot agree that in Table 1 the authors believe that 7.48 + - 0.04 is significantly different from 7.50 + - 0.12. Such inattention to one's own data is alarming.
